# Genera and Species of the Anisakidae Family and Their Geographical Distribution

**DOI:** 10.3390/ani10122374

**Published:** 2020-12-11

**Authors:** Juan C. Ángeles-Hernández, Fabian R. Gómez-de Anda, Nydia E. Reyes-Rodríguez, Vicente Vega-Sánchez, Patricia B. García-Reyna, Rafael G. Campos-Montiel, Norma L. Calderón-Apodaca, Celene Salgado-Miranda, Andrea P. Zepeda-Velázquez

**Affiliations:** 1Instituto de Ciencias Agropecuarias, Universidad Autónoma del Estado de Hidalgo, Rancho Universitario Av. Universidad km 1. Ex-Hda. de Aquetzalpa A.P. 32, Tulancingo 43600, Hidalgo, Mexico; juan_angeles@uaeh.edu.mx (J.C.Á.-H.); mvzfabiangomez@gmail.com (F.R.G.-d.A.); nydia_reyes@uaeh.edu.mx (N.E.R.-R.); vicente_vega11156@uaeh.edu.mx (V.V.-S.); patricia_garcia6857@uaeh.edu.mx (P.B.G.-R.); rcampos@uaeh.edu.mx (R.G.C.-M.); 2Departamento de Medicina and Zootecnia de Aves, Universidad Nacional Autónoma de Mexico, Av. Universidad 3000, Col. UNAM, C.U. Del. Coyoacán, Mexico City 04510, Mexico; nlca@unam.mx; 3Animal Health Research Center, Faculty of Veterinary Medicine and Animal Production, Autonomous University of the State of Mexico, Toluca 50295, Mexico; salgadomiranda@uaemex.mx

**Keywords:** anisakiasis, anisakidosis, Anisakidae, parasite, zoonotic, fish

## Abstract

**Simple Summary:**

The parasites of the Anisakidae family infest mainly marine mammals; however, they have the ability to infest paratenic hosts such as mollusks, small crustaceans and fish. The consumption of meat from animals of aquatic origin favors the acquisition of the disease known as Anisakiasis or Anisakidosis, depending on the species of the infecting parasite. Currently, the identification of the members of this family is carried out through the use of molecular tests, which brings about the generation of new information. The purpose of this review was to identify the genus and species of the Anisakidae family by reviewing scientific papers that used molecular tests to confirm the genus and species. The adaptability of the Anisakidae family to multiple hosts and environmental conditions allows it to have a worldwide distribution. As it is a zoonotic agent and causes non-specific clinical symptoms, it is important to know about the different members of the Anisakidae family, as well as the hosts where they have been collected.

**Abstract:**

Nematodes of the Anisakidae family have the ability to infest a wide variety of aquatic hosts during the development of their larval stages, mainly marine mammals, aquatic birds, such as pelicans, and freshwater fish, such crucian carp, these being the hosts where the life cycle is completed. The participation of intermediate hosts such as cephalopods, shrimp, crustaceans and marine fish, is an important part of this cycle. Due to morphological changes and updates to the genetic information of the different members of the family, the purpose of this review was to carry out a bibliographic search of the genus and species of the Anisakidae family identified by molecular tests, as well as the geographical area in which they were collected. The Anisakidae family is made up of eight different parasitic genera and 46 different species. Those of clinical importance to human health are highlighted: *Anisakis pegreffi*, *A*. *simplex*
*sensu stricto*, *Contracaecum*
*osculatum*, *Pseudoterranova azarazi*, *P*. *cattani*, *P*. *decipiens* and *P*. *krabbei*. The geographical distribution of these genera and species is located mainly in the European continent, Asia and South America, as well as in North and Central America and Australia. Based on the information collected from the Anisakidae family, it was determined that the geographical distribution is affected by different environmental factors, the host and the ability of the parasite itself to adapt. Its ability to adapt to the human organism has led to it being considered as a zoonotic agent. The disease in humans manifests nonspecifically, however the consumption of raw or semi-raw seafood is crucial information to link the presentation of the parasite with the disease. The use of morphological and molecular tests is of utmost importance for the correct diagnosis of the genus and species of the Anisakidae family.

## 1. Introduction

Transmission of diseases to humans caused mainly by the consumption of fish or fishery products is known as ichthyozoonosis [1]. These diseases are of bacterial, viral, fungal or parasitic etiology. Parasitic ichthyzoonoses are highly relevant, due to the severe clinical conditions that they can cause in humans [2].

One of the main families of parasites that have the ability to cause parasitic ichthyozoonosis is the Anisakidae family, which belongs to the phylum Nematoda and class Secernentea and the order Ascaridida, suborden Ascaridina, superfamily Ascaridoidea, family Anisakidae [3,4]. The nematodes of the Anisakidae family are distributed in a cosmopolitan way [5] and the family is made up of different genera, some of them with great zoonotic potential, such as the genus *Anisakis* [6,7]. For the development of the biological cycle of the parasite, the participation of marine fish is important, as they act as paratenic hosts or carriers of the L3 larval stage, where the larvae encyst or remain attached to the internal tissues [6]. The parasite has the ability to remain in the coelomic cavity or carry out larval migration to the epiaxial muscle of the infested fish; the life cycle is completed when the marine mammal and piscivorous birds consumes the paratenic host [8,9].

Acquisition of the parasite in humans occurs through the consumption of raw or semi-raw fish or marine products. Due to different existing cultural and gastronomic traditions, in the case of Mexico, the infection is acquired by the consumption of dishes such as aguachile or the popular ceviche [8], while in other countries, such as Japan, the consumption of dishes such as sushi and sashimi favor the presentation of parasitic ichthyozoonosis; in the Asian continent the average consumption of fishery products is 24 kg per year [10,11,12]. The clinical disease caused by these parasites is known as “anisakiasis”, when the infection is caused by the species *A. simplex sensu stricto* (s.s.), or “anisakidosis” when the infection is caused by *Contracaecum* spp. or *Pseudoterranova* spp. [13,14].

The clinical symptoms are nonspecific and may present as epigastric pain, nausea, vomiting, abdominal distention with intense pain and sometimes hypersensitivity reactions [13]. The development of molecular tests and genetic sequencing in the clinical field has allowed the correct identification of the different parasitic species related to clinical pictures, and that in turn allows understanding of the individually pathologies to which each of these genus are related [15]. On the other hand, the use of molecular tests has allowed the maintenance and/or discard of the parasitic genera that have been described morphologically, allowing clarification of which members really belong to the Anisakidae family. Based on the above, the purpose of this work was to carry out a bibliographic search of the genus and species that have been described in the Anisakidae family, identified by the use of morphological and molecular test; as well only through the use of molecular tests and the hosts that have been collected.

## 2. Life Cycle

Anisakarid nematodes have the ability to infest a wide variety of aquatic organisms during the development of their different stages, from egg to an adult capable of reproduction [16]. The adult stages are observed in the definitive hosts, such as marine mammals, among which are whales, belugas, dolphins, sea bears and seals [17,18,19,20,21,22,23,24,25,26]; as well as in different species of birds such as pelicans, penguins and herons [23,27,28,29,30,31,32,33,34,35,36].

To understand the life cycle of the parasite it is important to know that there are four larval stages, larvae 4 (L4) are males, and females. The females are capable of producing 1.5 million eggs. Oviposition increases in the last phase of life of the female, which is estimated to be at 30 to 60 days old [37]. The embryonated eggs are released into the intestine of the definitive hosts [38] and are eliminated through the feces to the aquatic environment, where the larval stages L1 and L2 develop [9]. It is important to mention that hatching does not occur in the digestive tract of marine mammals, due to temperature, since the action of other external requirements, such as low sea temperatures, salinity and the presence of oxygen favor the hatching of the egg [37]. In the sea, the egg with L2, will mix with plankton, krill, copepods and small crustaceans [38,39]. An important key of the Anisakidae family, to adapt and infest different hosts, radiate in the infestation of different organisms that are part of the trophic links of marine ecosystems from merozooplankton, like *Nyctiphanes couchii* (Euphausiacea) or *Sapa fusiformis* (Salpidae), just to mention a few; up to infest the top predators of importance for the life cycle [39]. The L2 uses these copepods to ingest it and release L2, L2 remains inside of these intermediate hosts until it reaches an optimal size for its molt to L3 [40], unless they are consumed by fish [41]. When marine fish consume the intermediate hosts, they act as paratenic hosts or carriers of stage L3 [9], which is trapped in the gastrointestinal tract and migrates toward the coelomic cavity; being free, it encysts in internal tissues such as liver, kidney and epiaxial muscle [42,43] or adheres to the serosa of internal tissues [6,8], causing an inflammatory response [9]. The life cycle of the parasite is completed when the fish infested with L3 is ingested by marine mammals where the larval stage L4, or the adult form, develops [9]. In birds, when they consume the infested fish, during the digestion process the L3 is released and the parasite is free to make its last molt, transforming into L4 and repeating the life cycle [40,44]. It is important to mention that L3 does not have the ability to develop to L4 in fish and humans, so parasitic reproduction in them does not take place [41] (Figure 1).

## 3. Overview of Morphology

Morphological identification was, and continues to be, the diagnostic tool for the general identification of the genera that make up the Anisakidae family [45]; however, it has been shown that only the visual examination is deficient [46]. Unfortunately, the morphology of the different parasitic genera that make up the Anisakidae family is not fully known, due to the development of the different larval stages that occur throughout the parasite’s life cycle [45,47]. Identification can be further complicated by morphological characteristics that are shared by different genera, such as relative size, special shapes, differences between males and females, as well as the nematode head and tail shape; and even the presence or absence of cuticular spicules on the body [48,49], as has been exposed by different authors (Figure 2).

Other morphological characteristics that have been described are those related to the eggs of the parasites, as in the case of *C. multipapillatum*, where its surface has been described as microvillous, which gives it a rough appearance when observed by scanning electron microscopy (SEM) [33]. In ascarid parasites, it has been suggested that the composition of the eggshell includes proteins and mucopolysaccharides, which give it a rough appearance, and with the presence of grooves or “opercular regions”, which serve as weak areas that allow the parasites to hatch [50]. The fibrous material of which the eggs are composed and the size of the embryonated eggs can vary from one genus to another, so the morphological characteristics may present variations that make identification difficult [33,41]. The differences in size, structure and surface of the eggs may also be involved with the total length of the parasite and the genus, so it would be necessary to carry out a more in-depth study of the different eggs in the different parasitic genus to establish a pattern between the different shapes, sizes, surfaces and aspects of embryonated eggs, in order to generate greater morphological knowledge of the Anisakidae family.

On the other hand, the basic characteristics that have been taken into account by different authors to establish morphological identification include the presence of the cuticular tooth, ventrolateral lips, excretory pore and its position, and the pointed shape of the head and tail, as well as the different measurements of total length, body diameter, esophagus length, nerve ring length, ventricle length, tail length and the presence of spicules, among others [35,38,51,52,53]. Based on the description and these morphological characteristics used to identify the nematodes of the Anisakidae family, it was determined that there is a total of 11 genera: *Anisakis* spp., *Contracaecum* spp., *Dujardinascaris* spp., *Goezia* spp., *Hysterothylacium* spp., *Mawsonascaris* spp., *Pseudoterranova* spp., *Phocascaris* spp., *Raphidascarididae* spp., *Sulcascaris* spp. and *Terranova* spp. However, based on genetic information, there are four genera that show some morphological similarities to the Anisakidae family, which have been discarded as a result of molecular studies.

The genus *Dujardinascaris* spp. has been described and collected mainly in cases of parasitosis in crocodiles and some species of fish in different parts of the world [53,54,55,56,57]. The presence of three semicircular lips, as well as the pointed shape of the head and tail can result in errors in identification, causing it be considered part of the Anisakidae family [58]; however, the morphology of this genus presents a row of teeth on each lip, as well as the presence of papillae throughout the entire body, with special emphasis on the pre-cloacal, pre-anal and post-anal areas [54].

*Dujardinascarias* spp. was discarded from the Anisakidae family after a phylogenetic study carried out by Mašová et al. [47], who used nematodes collected from the Nile crocodile (*Crocodylus niloticus*), where the generated clades were separated from the Anisakidae family but joined to the Heterocheilidae family.

Another parasitic genus identified as *Goezia* spp., has been collected from piraputanga (*Brycon hilarii*) [59], *Leporinus macrocephalus* (Anostomidae) [60], *Bagrus bayad* (Osteichthyes) [61] and *Tenualosa ilisha* (Cupleidae) [62], mainly parasitizing freshwater fish species. However, a study by Jackson et al. [63] identified in this nematode the presence of cuticular spines and flattened lips; these morphological characteristics are alien to the Anisakidae family, so it should not be grouped in it. A study by Silva et al. [64] reported through genetic analysis that the genus *Goezia* spp. belongs to the family Raphidascarididae, since the morphological characteristic of this family is the existence of an anal papilla present in males [65].

In a study by Shamsi et al. [66] on *Hysterothylacium* spp. parasites, obtained from different species of fish obtained from a local fish market, on the southeastern coast of the Iranian coast, molecular and morphological studies were carried out, the presence of papillum near to the labia, excretory pore at the height of the location of the nerve ring and the presence of spicules at the posterior ends were identified, however these characteristics have not been observed with other members of the Anisakidae family, but they have been described in the Raphidascarididae family.

Finally, the genus *Raphidascaris* spp. has been one of the genera with the greatest difficulty being separated from the Anisakidae family, since it shares some morphological characteristics, such as the presence of the boring tooth and the presentation of a ventricular appendix, which extends from the midbody [63]. In addition, since its inception it has been compared with the genus *Contracaecum* spp., due to similarities that occur in the shape of the intestine [45], as well as the presence of the three lips that the Anisakidae family also presents [65]. However, phylogenetic studies carried out on nematodes belonging to the genus *Raphidascaris* spp., in comparison to those belonging to the Anisakidae family, have shown that genetically these must be considered separately due to genetic analyses and the independent position of the clades generated in phylogenetic studies [65,66,67], although recent information can still be found where the genus *Raphidascarididae* spp. is still considered as part of the Anisakidae family [68].

## 4. Taxonomy Based on Molecular Evidence

Due to the morphological complexity of the nematodes that belong to the Anisakidae family and that in the best of cases only allows the identification of the parasitic genus [42], it has been necessary to choose the use of molecular studies, through which more precise identification can be made that cannot be achieved by morphological identification, and which in turn allows the correct identification of the members of the Anisakidae family to be clarified [65].

The molecular identification techniques of the parasite can vary depending on how the parasite was preserved, either in 70% alcohol, in glycerin with added phenol or included in paraffin for histopathological study [8,21]. However, for molecular tests, the extraction of DNA from the larva is of utmost importance; for this purpose, different commercial extraction kits can be used, with the protocols integrated by the company that offers them [69].

The identification of different morphological characteristics has generated a great advance in the identification of different genus and species of the Anisakidae family, although it can also complicate the identification of parasites that share certain morphological characteristics and similar life cycles, causing misdiagnosis, as well as that if the loss of some morphological characters occurs as the life cycle of the parasites progresses, such as the reduced size, size of the internal tissues, presence or not of different characteristics such as teeth, lips, pores, spicules, undulations of the cuticle, among others [70]. The implementation of these molecular tests is not only limited to determining new genus and species, but also allows the determination of the genetic information of the parasite, as well as the identification of the complete genome, genes and products of the genes that may be involved in the pathogenesis processes of a certain pathogen, in the virulence mechanisms and factors, without neglecting identification in the clinical diagnosis of the disease [71].

Some of the different molecular tests that have been used to identify members of the Anisakidae family, such as polymerase chain reaction (PCR), polymerase chain reaction-Restriction Fragment Length Polymorphism (PCR-RFLP) and next-generation sequencing (NGS), to mention a few [71,72,73]. The use of DNA barcoding has also had a great impact on identifying and discerning the different species of parasites, for this reason the use of molecular tests is an important complement for correct identification, without trying to displace the morphological identification [73], molecular tests try to complement the correct identification, by providing specific and potentially useful information in most cases for an early identification, either of the parasites themselves or of the vectors involved in the life cycle [70]. This complementation arises from the fact that the exclusive use of molecular tests can bring with it problems, since if the genetic information of the parasite is altered, likewise the results will also be altered. One of the best known examples is the use of markers molecular, as in the case of the 18S rRNA that allows to identify the 18S genes; however, it has been shown that the expression of genes is not the same in situations of isolation of the pathogen, such as gene expression in vivo, developed naturally during the life cycle of the parasite and its confrontation under different circumstances with the different hosts [70,73].

Thanks to the information provided by the various morphologic/molecular and molecular studies, it has been made clear that some genus should not be selected in the Anisakidae family; however, it has been identified that eight parasitic genera make up this family: *Anisakis* spp., *Contracaecum* spp., *Mawsonascaris* spp., *Phocascaris* spp., *Pseudoterranova* spp., *Pulchrascaris* spp., *Terranova* spp. and *Sulcascaris* spp.; these have been collected from different marine and aquatic hosts, among which are cetaceans, seals, belugas, dolphins, sea bears, sharks, fish, eels, manta rays, mollusks, penguins and pelicans (Table 1).

## 5. Geographical Distribution

The distribution of nematodes of the *Anisakidae* family is worldwide, and their identification through molecular tests has been carried out more frequently in the Asian continent [115], where the consumption of fish is higher, registering a per capita consumption of 24.1 kg/year [12], which is consistent with the Asian diet, which is based mainly on the consumption of raw and semi-raw seaFood In a study by Yokogawa and Yoshimura [116], they reported parasitic infestation by *Anisakis* spp. in Japanese patients with clinical symptoms affecting the gastrointestinal tract, using the term “larval anisakiasis” to determine that the disease was caused by a parasite of the Anisakidae family through the consumption of raw fish and squid. On the European continent, a per capita consumption of fish of 21.6 kg/year has been recorded, compared to Latin America and the Caribbean, where 10.5 kg/year is consumed. However, in the case of Oceania, the per capita consumption of fish is 24.2 kg/year, this being the highest registered worldwide [12]. The differences identified, based on consumption and scarce molecular identification of parasites of the Anisakidae family carried out in this continent, do not imply a lack of attention of the health sector in the face of ichthyzoonosis but may suggest that the presentation of the disease is common and that only treatment of the disease is important, without the need to identify the causative agent. Parasitic dissemination falls mainly on marine mammals [117], who are the definitive hosts of the Anisakidae family. Due to their eating habits, they are located near fishing areas where they consume the intermediate host, which affects their time of presentation of parasitosis in the different species of marine animals that are consumed by humans [118] (Table 1). The presentation of allergies caused by *A*. *simplex* s.s has been described more frequently in countries where the consumption of raw fish meat is a culinary tradition, such as in Japan, Peru and Mexico [119]. More than 90% of the zoonosis cases reported in humans have been in Japan, about 2000 cases/year being registered [120,121,122,123]. In Europe, 500 cases/year are reported, in Germany, France, Spain and the Netherlands [124,125,126] (Table 2, Figure 3).

The most frequent distribution areas of the Anisakidae family have been previously reported in the Mediterranean region, Japan region, North America and the North Atlantic Ocean region, since they are fishing areas of monetary importance [127]. Intentional or accidental discarding of fish or offal in fishing activities can favor the spread of the parasite [39]. Factors that favor the distribution of the Anisakidae family include the increase in temperature due to global warming, which has resulted in changes in latitude, changes in oceanographic conditions, as well as water circulation and salinity percentages [39,128]. In a study by Hojgaard [129], he identified that the survival time of *A*. *simplex* eggs increases due to high temperatures and high salinity.

Other factors that have also been linked to the distribution of parasites of the Anisakidae family in the parasite–host relationship are the land distances, temperature on the surface of the oceans and depth at which the hosts are found [127]. The thermal stress suffered by the hosts and that consequently promote their migration, while in the worst case this thermal impact can cause mortality in different marine species that could not adapt to changes in temperature [130]. These climatic changes have caused the migration and distribution of different species of marine animals in the oceans, which have resulted in different species being located in new geographical areas where they have not usually been reported [131]. This causes the dissemination of the invasive parasite in a new aquatic ecosystem and the infestation in new hosts [132], notably affecting the native species of a certain area as well as the presentation of the increased parasite prevalence and the presentation of parasitic co-infections [130,131]. On the other hand, the increased prevalence and parasitic density in paratenic hosts cause negative effects on weight gain, as in the cod (*Gadus morrhua*), due to the dissemination of the Anisakidae family caused by the gray seal (*Halichoerus grypus*) [101].

## 6. Anisakiosis/Anisakiasis

Parasitic ichthyozoonosis caused by some genus of the Anisakidae family can be acquired by humans through the consumption of different poorly cooked, semi-raw and raw marine animals, such as fish, squid, marine mollusks and octopus, among others [13]. Likewise, the different eating habits of a particular area and/or the acquisition of these habits in new geographical areas can result from the globalization of gastronomy [10]. The existence of culinary dishes where the fish is eaten raw or semi-raw includes sushi, sashimi, tuna tartare, herring and pickled anchovies; ceviche, tiradito smoked or salted herring; gravlax or thin, thin cuts of Scandinavian salmon meat; Spanish anchovies in vinegar (pickled anchovies); raw sardines, kuai, kokoda, kelaguen, fish tripe, among others [6,7,11,133].

The consumption of fish infested with the larval stage L3 causes the disease known as “anisakidosis”, which refers to the disease produced by *A. pegreffi* [78], *Contracaecum osculatum* [43], *Pseudoterranova azarazi* [134,135], *P*. *cattani* [26,136], *P*. *decipiens* [43] and *P*. *krabbei* [110]. While the term “anisakiosis” refers to the pathology caused specifically by the species *Anisakis simplex* s.s. [7,14,15,19,20,52,89]. One of the peculiarities of *A*. *simplex* s.s. is its ability to migrate more frequently to the epiaxial muscles of the fish, due to its high adaptability and physiological tolerance, which is why it tends to be more frequent in clinical cases in humans [82]; it also increases the risk of developing “anisakiosis” [13,14]. Likewise, it has been observed that infections caused by *P. decipiens* tend to be less invasive compared to those caused by *A*. *simplex* s.s., as well as the absence of severe gastric signs in reported cases for *P*. *cattani* [137] and *P*. *azarasi*, which have been collected from the palatine tonsillar [134]. When humans become accidental hosts, the parasite can survive for a short period of time without the ability to develop into adulthood and reproduce [138].

The first signs of the disease may present as a sensation of having something between the teeth, respiratory symptoms and nasal congestion, “tingling throat syndrome”, epigastralgia, nausea, gastric reflux, cough, dysphagia, vomiting and in some cases hematemesis has been reported due to gastric ulceration, depending on the location of the parasite [26,122,124,137,138].

The pathogenicity of anisakiosis/anisakiasis results from two main mechanisms: (1) direct tissue damage, and (2) an allergic response. Likewise, four categories have been determined in which the zoonotic species of the Anisakidae family can cause a clinical picture: (1) gastric, which occurs more frequently [126]; (2) intestinal [122]; (3) ectopic or extra-intestinal, where larval migration to the abdominal cavity occurs [122]. Likewise, there may be extraordinary cases where the parasite can encyst in the esophagus, causing eosinophilic esophagitis [121] and palatine tonsillar infection [134]; and (4) the generation of allergies [116].

Anisakiasis can be present as an asymptomatic, acute (or subacute) symptomatic or chronic symptomatic disease [118]. Once the body comes into contact with the parasite, different degrees of inflammatory responses develop, as well as changes in the permeability of blood vessels [139,140]. The body can react by developing granulomas in the submucosa, with abundant eosinophilic infiltrate (eosinophilic granuloma) and the presence of edema at the site of injury [8,141]. The survival capacity of the larva in humans is 2 to 3 weeks after being ingested [118,123].

It has been determined that the difference between gastric and intestinal presentation lies in the time since the raw shellfish was consumed, these being 7 and 36 h, respectively [123]. Some authors consider that the manifestation of the disease can occur in a period of 7 to 12 h after consumption of the parasite [6,137]. In approximately 95% of clinical cases, the larva can penetrate the gastric mucosa [136] and be observed with relative ease anchored and manifesting epigastric pain in the host until the larva dies or is surgically removed [142].

When the presentation is intestinal, it can cause abdominal distention and intense pain in the patient, which can be present for 5 to 7 days [13]. The presence of edema and abscesses in the mucosa and submucosa have been described, with a marked eosinophilic response around the larvae in the duodenum, jejunum [139] ileum and colon [143]. The main microscopic findings at the site of the injury are characterized by the presence of moderate to severe eosinophilic infiltrate and erosion of the mucosa [144]. Encystment has also been observed in the intestinal epithelium that can trigger the presentation of cancer; however, this will depend on the mutagenicity of the cells and the tumor-promoting potential of larval antigens [145]. In a study carried out by Murata et al. [115], a case of hepatic anisakidosis caused by *P*. *decipiens* was reported, which was first diagnosed in the patient as a neoplasm; however, when analyzing the tumor, it was identified that it was an eosinophilic granuloma, in which the larva was located at the center of the lesion.

The allergic response is caused by 28 allergens, among which is found proteins with antigenic roles, excretory secretory products and enzymes, that have been identified in *A*. *simplex* s.s. and *A*. *pegreffii* [71] among which are somatic and secretory antigens released by the larva when surgically removed, expelled by the body or when it dies within the body [141]. The main allergen recognized by the human body is the serum protein Ani s 1, identified in 87% of patients who develop the clinical picture, where previously fish infested with the nematode have been consumed [146]; Ani s 7 is an excretory antigen that has not been well characterized but is detected in 100% of patients with allergy caused by the parasite, during the acute phase of infection [71,147]; and Ani s 4, which is a cysteine detected in 27–30% of patients [147]. Many of these allergens are resistant to heat and/or pepsin; however, the most recognized by patients is the protein Ani s 1, a serine protease inhibitor that is heat stable and remains present even when the fish is cooked [148]. In the case of *A*. *pegreffii*, 2 antigenic proteins have been identified, which have the role of secretion products, being A.peg-1 and A.peg-13 which are expressed by the parasite when the temperature of the environment where the parasite ranges from 20 to 37 °C, respectively; for *Contracaecum rudolphii*, it has an enzymatic activity, among which are the activities of hydrolases that cause damage to the epithelium of the digestive tract, which allows the parasite to migrate freely in the host [71].

The clinical manifestations of an allergy can be moderate, when the presentation can manifest as irritation, inflammation, ulceration, secondary gingivostomatitis in the areas that have been in contact with the larva [149] and urticaria, which occurs in 60–70% of cases where there is a gastric presentation [7,150]; or severe, when a type I hypersensitivity reaction occurs, which can cause the presentation of angioedema, hypotension, bronchospasms, the presentation of anaphylactic shock and even asthma or worsening of a previous asthma, rhinoconjunctivitis, bronchoconstriction and dermatitis, without an acute infectious picture being present [125,140,141,142,148,151,152]. The presentation of the allergy caused by *Anisakis* spp. is more prevalent than other food allergies in adult humans, which includes 10% of idiopathic anaphylactic reactions [152].

It has been suggested that, to correlate the nonspecific clinical manifestations caused by Anisakidosis/anisakiasis, it is necessary to associate the information from the medical history, where information on the consumption of foods of marine origin, raw, semi-raw, pickled or other preparation [145]. However, due to the complexity of the clinical presentation, Anisakidosis/Anisakiasis are misdiagnosed, and can be confused with other conditions, such as appendicitis, gastric ulcer, tumors, cholecystitis, peritonitis, Crohn’s colitis, diverticulitis, intestinal intussusception, abdominal obstruction, appendicitis and peptic ulcers, also food allergies, bacterial and viral gastrointestinal infections; and even other intestinal parasites [106,121,133,153]. Some of the complications that can occur along with parasitic infection include intestinal obstruction, eosinophilic enteritis, spontaneous rupture of the spleen [154], presentation of eosinophilic granuloma, marked eosinophilia and abdominal distension [155].

Gastroscopy in the first stages of the disease allows to observe the mucosa and the presence of the parasite in situ, the use of computed tomography has been very useful to observe the area with the inflammatory process, which would suggest the presence of the parasite [123]. One of the cellular manifestations that have accompanied the infestation caused by *Anisakis* spp is the presentation of infiltrate and degranulation of the eosinophils, in the site where the parasite is found, this in turn also generates misdiagnosis problems, etiological agents such as *Baylisascaris procyonis*, *Gnathostoma* spp., *Echinocephalus* spp., among others can generate a similar cellular reaction [133,145,155], when the parasite is found in the intestinal tract, surgeries are performed to collect the parasite [156].

The ability of this parasite to adapt to different hosts allows the Anisakidae family to be collected in different sea animals and should be considered as a natural event throughout the life cycle of the Anisakidae family and not as a contaminating agent on fishery products [71]. As a natural event, it is important that humans know about its existence and the different ways of preventing infestation, through food safety and sanitary measures, as well as education for people about preventing disease [1]. In addition, very important medical staff must be updated on Anisakiosis/Anisakiasis to consider it as a possible diagnosis, using specific diagnostic tests that allow the correct identification of the etiological agent and, if the medical staff does not have experience with seafood parasites, approach an expert [129,133,157].

## 7. Conclusions

Based on the information collected from the Anisakidae family, it was determined that the geographical distribution is affected by different environmental factors, host factors and the parasite’s own ability to adapt and survive to external factors. Allowing it to be widely distributed worldwide. Likewise, its survival capacity and use of different pathogenicity factors and mechanisms influence the different phases of the life cycle through which the parasite goes through, thanks to this capacity for adaptation it has been considered a zoonotic agent. Although the clinical condition that the infection generates is not specific, it is important to suspect the presence of the parasite when the medical history shows the consumption of raw or semi-raw foods of marine origin. Likewise, the correct identification of the genera and parasitic species of the Anisakidae family, through the use of morphological and molecular tests, will help to have a more precise context and more information on the pathogenesis of the parasite.

## Figures and Tables

**Figure 1 animals-10-02374-f001:**
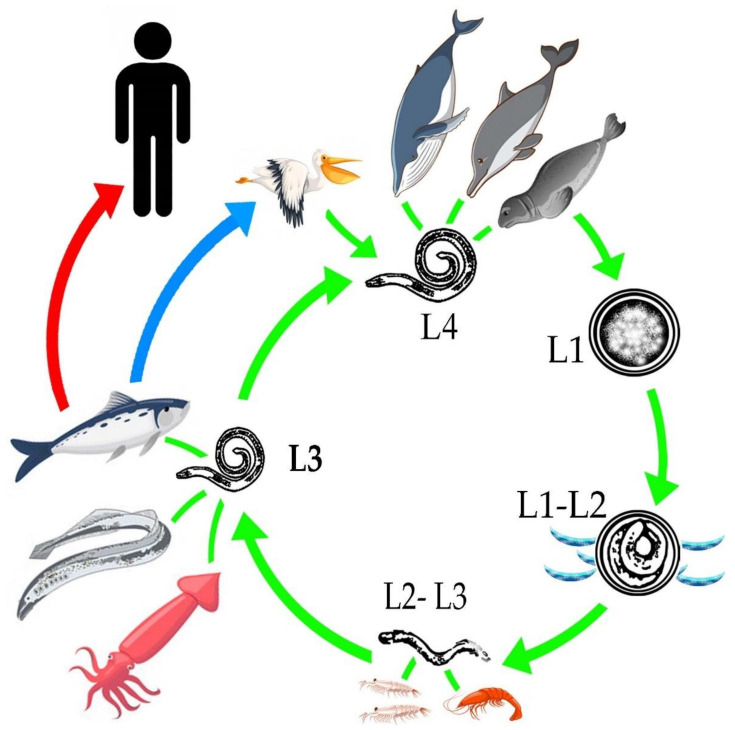
General life cycle of the Anisakidae family.

**Figure 2 animals-10-02374-f002:**
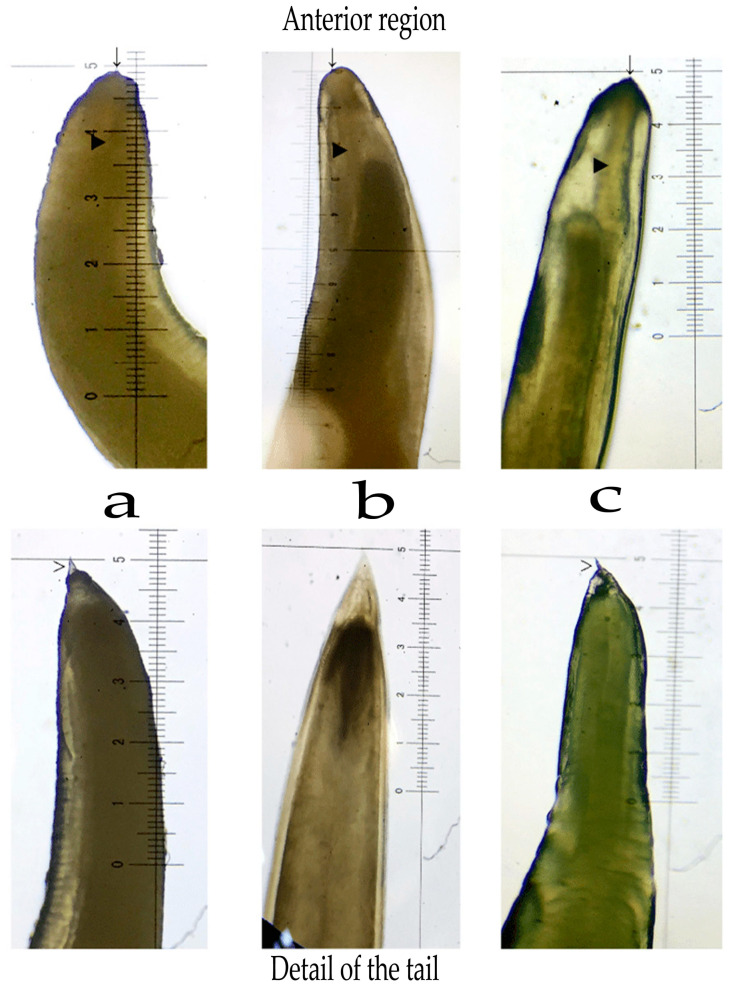
Morphological structures used for identification are observed. (**a**) *Anisakis* spp.; (**b**) *Contracaecum* spp.; and (**c**) *Psedoterranova* spp. [42].

**Figure 3 animals-10-02374-f003:**
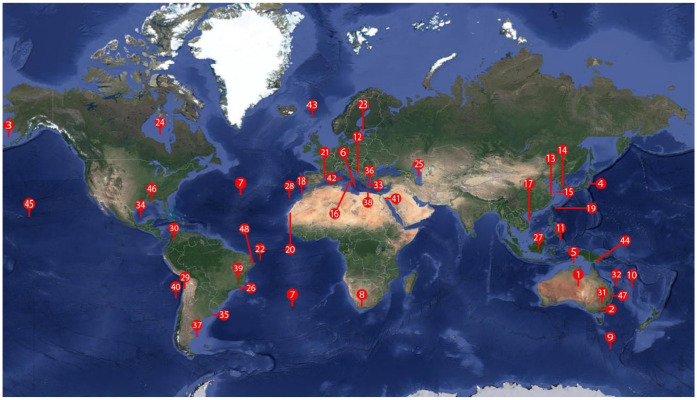
World map with geographical location of the parasites of the Anisakidae family, collected from different hosts.

**Table 1 animals-10-02374-t001:** Genus and species that make up the *Anisakidae* family, described and identified by morphology/molecular and molecular tests, isolated from various hosts.

Genus/Species	Host Common Name/Scientific Name	Map	Reference
*Anisakis berlandi*	Grey petrels/*Procellaria cinerea* Pygmy sperm whale/*Kogia breviceps*	1 2	[74,75]
	Pygmy sperm whale/*Kogia breviceps*	2	[75]
*A. brevispiculata*	Dwarf sperm whale/*Kogia sima* Pygmy sperm whale/*Kogia breviceps*	2 3	[18,75]
	Splendid Alfonsino/*Beryx splendens*	4 5	[4,76]
*A. nascettii*	European hake/*Merluccius merluccius* L.	6 7	[77]
	Black Scabbardfish/Aphanopus carbo	7	[78]
	Squid/*Moroteuthis ingens* Ziphiid species belong to the Genus/*Mesoplodon*	10 9 7	[79]
*A. paggiae*	Dwarf sperm whale/*Kogia sima*	3 2 10	[16,75,76]
	Pygmy sperm whale/*Kogia breviceps*	11 10	[17,76,80]
	Splendid Alfonsino/*Beryx splendens*	4	[4]
**** A. pegreffi*	Grey petrels/*Procellaria cinerea* Little penguin/*Eudyptulae minor*	1	[81]
	Sardine/*Sardina pilchardus* Anchovy/*Engraulidae* spp.	12	[81]
	Whitespotted conger/*Conger myriaster*	4	[82]
	Whitespotted conger/*Conger myriaster*	13	[83]
	Whitespotted conger/*Conger myriaster*	14	[84]
	Cinnamon flounder/*Pseudorhombus cinnamoneus*	15	[85]
	Sardine/*Sardina pilchardus* Silver scabbardfish/*Lepidopus caudatus* Electric lantern fish/*Electrona risso* Slender lightfish/*Vinciguerria attenuate* Spothead lantern fish/*Diaphus metopoclampus*	16	[86]
	Chum salmon/*Oncorhynchus keta*	14	[87,88]
	Redlip croaker/*Larimichthys polyactis* Atlantic mackerel/*Scomber scombrus* Largehead hairtail/*Trichiurus lepturus*	17	[89]
	Atlantic horse mackerel/*Trachurus trachurus*	18	[90]
	Largehead hairtail/*Trichiurus lepturus*	4 19	[91]
	Striped dolphin/*Stenella coeruleoalba*	20	[18]
	John Dory/*Zeus faber*	21	[34]
*A. physeteris*	Common Atlantic grenadier/*Nezumia aequalis* Roughsnout grenadier/*Trachynicus scabrus*	22	[52]
	Common Atlantic grenadier/*Nezumia aequalis* Roughsnout grenadier/*Trachynicus scabrus*	16	[88]
	Splendid Alfonsino/*Beryx splendens*	4	[4]
**** A. simplex* s.s.	Redlip croaker/*Larimichthys polyactis* Atlantic mackerel/*Scomber scombrus* Largehead hairtail/*Trichurius lepturus*	18	[89]
	Northern fur seal/*Callorhinus ursinus*	23 24	[19]
	Beluga whale/*Delphinapterus leucas*	25	[20]
	Common Atlantic grenadier/*Nezumia aequalis* Roughsnout grenadier/*Trachynicus scabrus*	23 25	[52]
	Chum salmon/*Oncorhynchus keta*	3 21	[82]
	Largehead hairtail/*Trichiurus lepturus*	3 21	[91]
	Brown shrimp/*Crangon crangon*	26	[92]
*A. schupakovi*	Caspian seal/*Phoca caspica*	27	[21]
*A. typica*	Sandperch/*Pseudopercis semifasciata* Namorado sandperch/*Pseudopercis numida* Brazilian sandperch/*Pinguipes brasiliansus* Patagonian flounder/*Paralichtys patagonicus* Fantail flounder/*Xystreurys rasile* Atlantic spotted dolphin/*Stenella frontalis* Fraser’s dolphin/*Lagenodeplhis hosei*	28	[16,22,93]
	Bullet mackerel/*Auxis rochei rochei*	29	[94]
	Largehead hairtail/*Trichiurus lepturus*	4 21	[91]
	Longnose trevally/*Caragoides chysophrys* Bumpnose trevally/*Carangoides hedlandensis* Imposter Jack or Imposter Trevally/*Carangoides cf talamparoides* Bigeye scad/*Selar crumenophtalmus* Dorab wolf-herring/*Chirocentrus dorab* Giant herring/*Elops cf hawaiensi* Ornate Snapper/*Pristipomoides argyrogrammicus* Silver moonyfish/*Monodactylus argenteus* Jpanese whiptail/*Pentapodus nagasakiensis* Indian mackerel/*Rastrelliger kanagurta* Orange-spotted grouper/*Epinephelus coioides* Bigeye barracuda/*Sphyraena forsteri* Sawtooth barracuda/*Sphyraena putnamae* Brushtooth lizardfish/*Saurida undosquamis* Live sharksucker/*Echeneis naucrates* Areolate grouper/*Epinephelus areolatus* Threadfin bream/*Nemipterus furcosus* Bartailed goatfish/*Upeneus vittatus* Mackerel scad/*Decapterus macarellus Scomberoides* sp.	10	[76,95]
	Spinner shark/*Carcharhinus brevipinna* Blue-lipped sea krait/*Laticauda laticaudata*	10	[74]
*A. ziphidarum*	Chub mackerel/*Scomber japonicus*	29 30	[21,96]
	Atlantic grenadier/*Nezumia aequalis* Roughsnout grenadier/*Trachynicus scabrus*	23	[52]
	Electric lantern fish/*Electrona risso* Spothead lantern fish/*Diaphus metopoclampus* Slender lightfish/*Vinciguerria attenuata*	16	[88]
	Cetaceans	28	[16]
	Pygmy sperm whale/*Kogia breviceps*	11	[17]
	Black-scarbbard fish/*Aphanopus carbo*	29	[96]
*Contracaecum australe*	Olivaceous cormorant/*Phalacrocorax brasilianus*	1	[97]
*C. bancrofti*	Great white pelican/*Pelecanus onocrotalus*	31	[23]
	Common carp/*Cyprinus carpio* Carp gudgeon/*Hypseleotris* sp Gambusia/*Gambusia holbrooki* Weather loach/*Misgurnus anguilicaudatus* Rainbow fish/*Melanotaenia fluviatilis* Bony bream/*Nematalosa erebi* Australian smelt/*Retropinna semoni*	32	[98]
*C. bioccai*	American brown pelican/*Pelecanus occidentalis*	33	[29]
*C. chubutensis*	Olivaceous cormorant/*Phalacrocorax brasilianus*	1	[97]
*C. eudyptulae*	Little penguin/*Eudyptula minor*	31	[23]
*C. fagerholmi* n.	American brown pelican/*Pelecanus occidentalis*	34	[30]
*C. galeocerdonis*	Elasmobranch fish	35	[22]
*C. gibsoni*	Dalmatian pelican/*Pelecanus crispus*	36	[31]
*C. margolisi*	Mammals of the pinniped family	31	[23]
*C. mirounga*	Magellanic penguin/*Spheniscus Magellanicus*	37	[32]
*C. microcephalum*	Australian pelican/*Pelecanus melanoleucos*	31	[23]
	Pelicans/*Pelecanus* spp.	31	[23]
*C. multipapillatum*	American brown pelican/*Pelecanus occidentalis*	34	[33]
*C. ogmorhini*	Australian fur seal/*Arctocephalus pusillus doriferus* New Zealand fur seal/*Arctocephalus fosteri*	31	[23]
	Australian pilchard/*Sardinops sagax* Aanchovy/*Engraulidae* sp.	32	[99]
**** C. osculatum*	Northern fur seal/*Callorhinus ursinus*	24 25	[19,20]
	Baltic cod/*Gadus morhua* Grey seal/*Halichoerus grypus*	26	[100,101]
	Amphipod crustacean/*Gammarus* spp.	26	[102]
*C. overstreeti*	Flathead grey mullet/*Mugil cephalus* Dalmatian pelican/*Pelecanus crispus*	38	[102]
	Pelicans/*Pelecanus* spp.	36	[31]
*C. pelagicum*	Magellanic penguin/*Spheniscus magellanicus*	39	[35]
	Magellanic penguin/*Spheniscus magellanicus*	3	[36]
*C. rudolphii* A, B, C, D and E	Neotropic cormorant/*Phalacrocorax brasilianus* Eel/*Anguilla Anguilla* European seabass/*Dicentrarchus labrax* Mediterranean banded killifish/*Aphanius fasciatus* Big-scale sand smelt/*Leuciscus cephalus* Barbel/*Barbus barbus* Crucian carp/*Carassius carassius*	1	[103]
*C. rudolphii* D and E	Black cormorant/*Phalacrocorax carbo* Pied cormorant/*Phalacrocorax varius*	1	[104]
*C. pyripapillatum*	Australian pelican/*Pelecanus conspicillatus*	2	[105]
*C. rudolphii* F	American brown pelican/*Pelecanus occidentalis*	34	[30]
*C. septentrionale*	Neotropic cormorant/*Phalacrocorax brasilianus*	1	[97]
*C. variegatum*	Australian darter/*Anhinga melanogaster* Australian pelican/*Pelecanus conspicillatus*	31	[23]
*Mawsonascaris australis*	Brown guitarfish/*Rhinobatos schlegelii*	4	[106]
*M. vulvolacinata*	Cowtail stingray/*Pastinachus sephen*	35	[107]
*Phocascaris crystophorae*	Northern fur seal/*Callorhinus ursinus*	25	[17]
**** Pseudoterranova azarazi*	Northern fur seal/*Callorhinus ursinus*	25	[19]
	Steller’s sea lion/*Eumetopias jubatus* Californian sea lion/*Zalophus californianus* Harbor seal/*Phoca vitulina richardsii* Bearded seal/*Erignathus barbatus*	11	[22]
*P. bulbosa*	White whale/*Delphinapterus leucas*	24	[20]
	Bearded seal/*Erignathus barbatus*	4	[22]
**** P. cattani*	South American sea lion/*Otaria flavescens* (*Otaria byronia*)	40	[25]
	South American sea lion/*Otaria flavescens* (*Otaria byronia*)	1	[26]
	South American sea lion/*Otaria flavescens* (*Otaria byronia*)	41	[22]
**** P. decipiens* (Sensu stricto)	Atlantic cod/*Gadus morhua* Pacific cod/*Gadus macrocephalus*	42	[108]
	Spinner dolphin/*Stenella longirostris*	28	[16]
	White whale/*Delphinapterus leucas*	24	[20]
	Pink ear emperor/*Lethrinus lentjan*	7	[109]
	Californian sea lion/*Zalophus californianus* Harbor seal/*Phoca vitulina richardsii* Harbor seal/*Phoca vituline* Grey seal/*Halichoerus grypus* Hooded seal/*Cystophora cristata* Norhern elephant seal/*Mirounga angustirostris*	7	[22,109]
**** P. krabbei*	Atlantic cod/*Gadus morhua* Atlantic horse mackerel/*Trachurus trachurus*	43	[110,111]
	Harbor seal/*Phoca vituline* Grey seal/*Halichoerus grypus*	7 44	[22]
*Pulchrascaris australis *n. sp.	Scalloped hammerhead/*Sphyrna lewini*	5	[22]
*P. chiloscyllii*	Brownbanded bambooshark/*Chiloscyllium punctatum* Blacktip reef shark/*Carcharinus melanopterus* Gummy shark/*Mustelus antarcticus* Scalloped hammerhead/*Sphyrna lewini* Smooth hammerhead/*Sphyrna zygaena* Whitetip reef shark/*Triaenenodon obesus*	8 45 46	[22]
*Terranova caballeroi*	Green water snake/*Nerodia cyclopion*	47	[112]
*T. galeocerdonis*	Sand tiger shark/*Carcharias taurus*	37	[27,28]
	Redbelly yellowtail fusilier/*Caesio cuning* Lowly trevally/*Caranx ignobilis* Shark mackerel/*Grammatorcynus bicarinatus* Mangrove red snapper/*Lutjanus argentimaculatus* Stripey sea perch/*Lutjanus carponotatus*	35	[22]
	Tiger shark/*Galeocerdo cuvier* Scalloped hammerhead/*Sphyrna lewini* Smooth hammerhead/*S. zygaena* Blacktai reef shark/*Carcharinus amblyrhynchos*	43 48 49	[22]
*T. pectinolabiata*	Great hammerhead shark/*Sphyrna mokarran*	32 31	[113]
*Sulcascaris sulcata*	Bivalve mollusks	20	[111]
	Mediterranean mussel/*Mytilus galloprovincialis*	43	[114]

*** Zoonotic species of *Anisakidae* family, reported on clinical cases.

**Table 2 animals-10-02374-t002:** Geographical localization of the *Anisakidae* family, identified by molecular tests in different countries and continents around the world.

Map	Localization/Continent	Map	Localization/Continent	Map	Localization/Continent
1	Australia/Oceania	18	North of Morocco/Africa	35	Coast of Argentina/South America
2	Victoria Australia/Oceania	19	Taiwanese waters/Asia	36	Aegean Sea, Turkey/Europe and Asia
3	St. Paul Island, Alaska/North America	20	Senegal/Africa	37	Argentina, Sea of Patagonia/South America
4	Japanese waters/Asia	21	Mediterranean coast, Spain/Europe and Africa	38	Alexandria City, Mediterranean Sea, Egypt/Africa
5	Water Australia/Oceania	22	Brazilian waters/South America	39	Minas Gerais/South America
6	Mediterranean sea/Europe, Africa and Asia	23	Baltic Sea/Europe	40	South east Chilean coast/South America
7	Atlantic Ocean	24	Hudson Bay and Hudson Strait, Canada/North America	41	Hurghada City, Gulf of Suez, Red Sea, Egypt/Africa
8	South Africa/African	25	Caspian Sea/between Europe and Asia	42	Tyrrhenian coast of southern Italy/Europe
9	Macquarie Island, Pacific Ocean southwest/Asia and Oceania	26	Rio de Janeiro, Brazil/South America	43	Faeroe Islands/ Europe
10	New Caledonia/Oceania	27	Indonesia/Asia	44	Halfway Island, Australia/Oceania
11	Philippine archipelago/Asia	28	Madeiran waters Portugal/African	45	Hawaii/North America
12	Northern Adriatic Sea/Europe	29	Chile/South America	46	Louisiana, USA/North America
13	Zhoushan, Zhejiang, China/Asia	30	Colombia/South America	47	Twynams Paar, Ceylon, South Australia and Queensland/Oceania
14	Republic of Korea/Asia	31	Southern New South Wales, Australia/Oceania	48	Natal, northern, Brazil/South America
15	Yellow Sea/Asia	32	Heron Island, Queensland. Australia/Oceania		
16	Sicily and Messina/Italy/Europe	33	Greek waters/Europe		
17	China Sea/Asia	34	Northern Gulf of Mexico/Central America

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
