# Peer review of "Genera and Species of the Anisakidae Family and Their Geographical Distribution"

_animals, 2020, doi:10.3390/ani10122374_

Round 1
Reviewer 1 Report
The review by Ángeles-Hernández et al. has summarized the studies related to the significance and geographic distribution of parasites belonging to Anisakidae family and their geographical distribution. This review manuscript is well written and very informative. However, there are a few broad recommendations, which might help to reach a broader audience.
- Please provide a generic illustration of the life-cycle of Anisakides.
- Please discuss the importance of DNA barcoding for the characterization and identification of Anisakides.
- The authors can provide pictures of selected Anisakid species.
- The authors should discuss the effect of climate change / rising water temperature on the geographical distribution of Anisakids.
- Are there any other factors which affect the geographical distribution of Anisakids, like birds/ migratory fishes/ water currents, etc. – please discuss in the manuscript.
Author Response
Dear reviewer:
The scientific team made the pertinent corrections, based on what you indicated.
New paragraphs were included, where information corresponding to the importance of molecule testing and how DNA barcoding helps the identification question was included.
2 figures were annexed, one of them representing some morphological characteristics considered for morphological identification (from various authors cited) and another figure that exemplifies the general life cycle of Anisakarids (from our authorship), finally we include the information on how it affects climate change and variables that affect the distribution of the Anisakidae family.
Thank you very much for your review.
All the scientific team send you greetings.
Reviewer 2 Report
The present review is of interest for people who works in parasitology and has interest in fish-borne zoonotic diseases. General aspects are of interest, but sometimes not deepen. My advice is to resubmit the review with some modifications, as reported below:
Line 18: Anisakiasis instead of Anisakiosis if you mean the human disease (the use of osis is typical terminology of veterinary infections). Also at line 59.
Line 20-22: “The purpose of this review was to identify the genera and species of the Anisakidae family by molecular tests. Based on the information collected, it was concluded that the Anisakidae family consists of eight parasitic genera and 47 species.” It is not clear to me if you performed a review of the molecular methods used, or if you performed molecular studies themselves. This issue is related also to lines 66-68, in which is not very clear who decide to retain or include some species or genera in the Anisakidae family. I suppose that you made an inventory of molecular studies, but aspects related on criteria you used to include or remove a particular study should be better described.
Line 25: please cite that are the adults those living in final hosts as marine mammals and aquatic birds. Moreover, freshwaters are included as well in the life cycles of some species, so please cite it.
Line 32 Anisakis peggreffi to be removed and substituted with Anisakis pegreffii, as in the remaining text. Moreover A. simplex is not the species name but the name of the species complex, i suppose you meant A. simplex sensu stricto.
Linr 46 the order is not Ascarida but Rhabditida. Ascaridoidea is the superfamily.
Line 65 typo error genres
Line 75 delete mainly as adults are observed only in final hosts.
Line 76: are you sure about the presence of skarks among final hosts?
Section 3 – Please tell in this section that morphology allow to identify genera and species if it is performed in adults, but adults are not easily recovered and usually are larval forms that need to be identified. This is the aspect that limit the easy identification of such species, together with morphological convergence of larval forms even belonging to different genera. Then, you should consider to deal with Hysterothylacium (Raphidascarididae family) in a different way: it was originally included in Anisakidae family, but later studies included in Raphidascarididae family, that the authors considered erroneously as a genus (line 148). Please correct it.
Anisakis nascettii is missing from the table. However, how did you select hosts for the Table? Because several reports are missing, so I was wondering which was the selecting criteria.
Line 267: please update the list of allergens identified in Anisakids. You can refer to the recently published review by D’Amelio and colleagues. (D’Amelio, S.; Lombardo, F.; Pizzarelli, A.; Bellini, I.; Cavallero, S. Advances in Omic Studies Drive Discoveries in the Biology of Anisakid Nematodes. Genes 2020, 11, 801.).
Please remove lines 293-297 which are redundants and are not your inferences.
Author Response
Dear reviewer:
The scientific team made the pertinent corrections, based on what you indicated.
The minor corrections you mentioned were made, such as the ending "osis" to refer to the disease in humans. We inform you that table 1, which contains the genera and species, has been modified. Some were genera were eliminated and others were annexed.
In the document we also indicate that our selection criteria were based on: works with morphological tests and molecular tests and molecular works.
The "shark" was eliminated as the definitive host, freshwater species were annexed. The information corresponding to allergens was updated and the appointment that you suggested was included. In the document you can find other elements that were included.
We appreciate your valuable time.
All the scientific team send you greetings.
Reviewer 3 Report
Thank you for inviting me to review the manuscript “Genus and species of Anisakidae family and their geographical distribution”. There are too many issues to be addressed. I have only listed the main ones here. This year alone I reviewed over 50 papers about anisakid nematodes for various journals and I usually recommend “major revision” and then "accept" if authors are ready to take major and extensive revisions of their works, if necessary, because I think a reviewer’s job is also to coach and guide new researchers to the field. But this manuscript as it stands has no leg to be recommended for publication. I hope authors do not see this as disheartening, instead I hope my comments helps them to change this work to something novel for science.
The entire information is already publically available through previous publications such as those by Mattiucci or in a number of books and also in WoRMS database so the manuscript really adds nothing new to the current knowledge.
I’m afraid this manuscript also contains several mistakes and misinterpretations of previous findings by leading authors publishing about Anisakid nematodes which not only damage authors’ reputation but also will not be good for the journal’s reputation. I have listed some examples at the bottom of this letter for authors to elaborate what I mean.
In addition numerous important publications have been ignored. Some have been listed below.
When I read Lines 54-56 where authors wrote: “Due to different existing cultural and gastronomic traditions, in the case of Mexico the infection is acquired by the consumption of dishes such as aguachile or the popular ceviche [8]”, it reminded me that knowledge about these important parasites in Mexico and central Nearctic region is really limited, particularly that many early publications are in non-English language. I would like to strongly encourage the authors that instead of poorly rewriting the available literature, as they did in the current manuscript, to turn their focus on writing about these parasites in a new setting that not much is known about. An article about anisakid specifically about a large region like Mexico or adjacent countries would be new to science, in particular if they can discuss how occurrence and ecology of anisakids in Mexico is different from the way these parasites behave in the rest of the world. The current manuscript won’t have any benefit for authors to establish a name among other researchers in the field, but a manuscript on anisakids in Mexico and comparing with the occurrence elsewhere in the world will make them to be known as champions of these parasites in their region, such as S Mattiucci in in Europe, S Shamsi is in Australia or L Zhang and L Li in China, and will help them to build a respectful profile for their future.
Here I provide some examples of areas that are erroneous, misinterpreted and in need of serious attention:
Examples from the Introduction
- Line 52: Please note that marine mammals are not the only definitive hosts for Anisakidae.
- Line 46: Order Asacrida is not right. Please make yourself familiar with the latest accepted taxonomy and the spelling of these parasites.
- Lines 68 to 70: Authors wrote: “the purpose of this work was to carry out a bibliographic search of the genera and species that have been described in the Anisakidae family and identified through the use of molecular tests, as well as the hosts that have been collected.” but the rest of the manuscript does not follow this aim at all. Firstly what type of molecular tests do you mean? MEE as established by Mattiucci et al? combined morphology and DNA sequencing as established by Shamsi et al? Plus Identification based on molecular tests alone can be erroneous as well which has resulted in many mislabelled sequences in the GenBank, such as sequences deposited by Jaiswal and others. Similarly morphology alone of larvae and females are not reliable. The authors must specify what criteria they had for quality control of molecular identification of larvae before their inclusion in their work? Some guidelines have been published by various authors such as Fagerholm, Nadler, Mattiucci, Paggi and Shamsi that they should have considered.
Examples from the Life cycle:
- Line 75 to 76: sharks have been listed among examples of marine mammals!
- Line 75 to 76: many important groups of animals that can be definitive hosts of the family have been overlooked.
- Lines 78 to 79: This sentence does not make any sense in regard to the developmental stage and again it would be detrimental for authors’ reputation if they publish an article containing statements like this, as one sentence by itself contains several mistakes. There are several excellent works on life cycle of these parasites and various developmental stages by Russian authors, Fagerholm, and many others that they can refer to. Please make sure you really understand what the authors are trying to say in their papers. As the manuscript has been written it feels like statements have been copied without any critical thinking behind.
- Line 80: size of females again is not correct. It is about anisakids as family then a range must be provided as the size is significantly different. Why size of females only and not males? Especially that females really hold no value in the identification of the groups. Also not sure how size of female is relevant to be mentioned in this section, under the life cycle? Also why unusual hosts for the life cycle of the parasites as published in Parasitology International, 2017, 66(6), 837-840 or the comprehensive review article by Mattiucci published in Advances in Parasitology have not been cited here?
- There are many more issues in this section. Really too many to list them all.
Overview of morphology:
Some examples of major issues include:
- Line 121: Excretory pore by itself does not have any value in identification; it is the position of the excretory pore and also the anatomy of the excretory system that is highly valuable identification tool.
- Lines 125 to 127: Are authors sure they disagree with existing Anisakid taxonomists, such as Nadler, Fagerholm, Shamsi, Mattiucci, Paggi, Li, Zhang and else! Seriously with all those evidence out there do you still place Hysterothylacium, Goezia and some of the others among family Anisakidae? What evidence do you have to justify your selection? All published articles by senior taxonomists in 2018 to 2020 placed for example Hysterothylacium spp under Raphidascarididae and so on. Authors did not even bother to provide any justifications to disagree with the existing classification. I wonder if they have seen and understood the phylogenetic trees or made for Hysterothylacium spp? That clearly shows they are way different from Anisakidae altogether. I understand the taxonomy of these parasites is confusing and that’s why I am surprised not see an experienced parasitologists who has worked on these parasites among the authorship? It is really important for them in the future work and at the time of revising this work to have help from a senior and experienced person as they cited some relevant papers but clearly they could not interpret them correctly!! Anyhow, please revise the list of genera. Some accepted genera for the genera occurring in Australia can be found in a review paper by Shamsi in International Journal for Parasitology: parasites and Wildlife, 2014 that can be a starting point to help authors.
- Lines 130 to 158: Delete as none of these are Anisakidae!
- Line 170 to 176: Please see the comments above about what is among Anisakidae and what isn’t!
- Table 1: Some of the important species such as Anisakis berlandi also known as Anisakis simplex C, Contracaecum pyripillatum, and many more are missing!
- Table 1: 78 is not the right reference for rudolphii D and E.
- Table 1: According to the statement in the introduction on the purpose of this study, it seems that authors tried to include both intermediate and definitive hosts as long as molecular data exist. But they did not include examples like C. bancrofti in carp and other fish? see publications in J Helmitol in 2017 and 2019.
- Table 1: Again despite Authors claiming the purpose of the study is to carry out a work on species that identified using molecular tests but then in their table they reuse to specify 5 to 6 different C. osculatum!
- Table 1: what is chiloscyiti? As far as I know there is no valid species as C. chiloscyiti. Authors made another mistak here by citing Shamsi and Suthar. They (Shamsi and Suthar) reported Terranova larval type which later were identified as larval stages of T. pectinilabiata and P. australis. Have you read these papers and paid attention to the phylogenetic trees that authors presented? Parasitology Research, 2020, 119(6), 1729-1742 and Parasitology Research, 2019, 118(7), 2159–2168
- Table 1: remove all Hysterothylacium spp from the Table.
- Table 1: you used wrong reference for Mawsonascaris. The correct one is Journal of Fish Diseases, 2019, 42(7), 1047-1056.
- Table 1; why members of Pulchrascaris have not been included?
Geographical distribution
- Please rewrite the entire section after you address all the inclusion and exclusion of members of Anisakids correctly.
Generalities of anisakiosis/anisakiasis
- The word “generalities” should be removed.
- Also in this section please include issues with misdiagnosis. You may find these articles informative as a start:
Roser, D., & Stensvold, C. R. (2013). Anisakiasis mistaken for dientamoebiasis? Clinical Infectious Diseases, 57(10), 1500-1500. https://doi.org/10.1093/cid/cit543
Sakanari, J.A. Anisakis—From the platter to the microfuge. Parasitol. Today 1990, 6, 323–327.
Shamsi, S., & Sheorey, H. (2018). Seafood‐borne parasitic diseases in Australia: are they rare or underdiagnosed? Internal Medicine Journal, 48(5), 591-596.
Shamsi, S. (2019). Seafood-Borne Parasitic Diseases: A “One-Health” Approach Is Needed. Fishes, 4(1), 9. https://www.mdpi.com/2410-3888/4/1/9
Examples of Minor comments:
- Line 41: Please provide a genuine reference for the word “ichthyozoonosis”
- Line 47 and 48: “such as the genus Anisakis spp”. In this context you should remove “spp”
- Line 59: Please pay attention to the spelling of “anisakiosis”
- Table 1: Fagerholmi n. should be C. fagerholmi
- Line 225: scientific names should be italic
Author Response
Dear reviewer:
The scientific team made the corrections, based on what you indicated.
The major corrections were made regarding the genera and species, we undertook the task of verifying and reading the citations that you indicated, which were attached to the document. The information on the life cycle of the parasite was corrected, we eliminated the size information of the females. The "shark" was eliminated as the definitive host.
The table that talks about the genus and species was modified, freshwater species were added, genus were eliminated and new ones were sought. We correct the miscited references, we add new information about the help of molecular tests in conjunction with the molecular ones, for a correct identification.
Table two was corrected, about the locations and the map that shows them. the misdiagnosis part was annexed and the suggested references were included.
We appreciate your valuable time and we hope to meet your expectations.
All the scientific team send you greetings.
Round 2
Reviewer 3 Report
Line 22: there are two full stops
Line 22 to 23: “Based on the information collected, it was concluded that the Anisakidae family consists of eight parasitic genera and 47 species.” this is basically not correct, plus it is already known for example through World register of Marie species. I suggest you use something novel here. You may like to say:
Line 47: the use of “Shamsi 202” after the name of the family is taxonomically wrong. If must use it as numbered citation like the rest of the citations.
Line 82: “where larvae 4 (L4) are sexually active adult males and females.” It is not possible that larvae to be sexually active!
Line 103: change “change” to “develop”
Figure 1: there are two many wrong arrows in that figure!
Line 172: “the genus Raphidascarididae spp.” is taxonomically wrong
table 1: still many reports are missing, for example I could not see these publications:
Increasing occurrence of anisakid nematodes in the liver of cod (Gadus morhua) from the Baltic Sea: Does infection affect the condition and mortality of fish? Fisheries Research.2016. 179, 98-103.
Occurrence of anisakid parasites in marine fishes and whales off New Caledonia. Parasitology Research, 2018, 3195-3204.
Characterisation of Ascaridoid larvae from marine fish off New Caledonia, with description of new Hysterothylacium larval types XIII and XIV. Parasitology International. 2015, 64, 397-404.
Zoonotic nematode parasites infecting selected edible fish in New South Wales, Australia. International Journal of Food Microbiology. 2019.
Anisakis Dujardin, 1845 infection (Nematoda: Anisakidae) in Pygmy Sperm Whale Kogia breviceps Blainville, 1838 from west Pacific region off the coast of Philippine archipelago. Parasitology Research. 2016, 115, 3663-3668.
Occurrence of Anisakis spp. (Nematoda: Anisakidae) in a pygmy sperm whale Kogia breviceps (Cetacea: Kogiidae) in Australian waters. Diseases of Aquatic Organisms. 2019, 134, 65-74.
Author Response
Dear reviewer:
The scientific staff inform you that minor revisions were made, as well as the inclusion of paragraphs in the section of conclusions and modification in the abstract.
The life cycle figure was also edited, as was the table of genus and species, locations and all bibliographic references, since they included the works that he kindly indicated to us.
We appreciate your valuable time and send you regards